

# R-Package `BIOdry`: DendroClimatic Modeling from Multilevel Ecological Data Series

Wilson Lara[1,2], Stella Bogino[3], and Felipe Bravo[1]

[1]Sustainable Forest Management Research Institute, UVA-INIA, Avda. Madrid, s/n, 34071, Palencia, Spain
[2]Research Center on Ecosystems and Global Change, Carbono & Bosques (C&B), Calle 51A, $N^o$ 72-23, Int: 601, 050034, Medellín, Colombia
[3]Departamento de Ciencias Agropecuarias, Universidad Nacional de San Luis, Avenida 25 de Mayo 384, 5730, Villa Mercedes, San Luis, Argentina

*Correspondence to:* Wilson Lara, wilson.lara@alumnos.uva.es, wilarhen@gmail.com

**Abstract.** R-package `BIOdry` is developed to consider ecological factors in dendroclimatic modeling of forest ecosystems. The package processes Multilevel Ecological Data Series (MEDS) with two functions: `modelFrame()` and `muleMan()`. These integrate techniques for modeling Tree Rings in Wood (TRW) chronologies and climatic series at specific ecological-factor levels (one-level modeling), developing multilevel analysis (mixed-effects detrending), and implementing multivariate

comparison of MEDS. Several one-level functions for serial synchronization, drought modeling and allometric scaling are implemented to model tree-diameter growth and drought of Iberian pine forests (*Pinus pinaster* Ait.). Trends in the dendroclimatic MEDS are subtracted with in-package detrending methods. Extracted fluctuations are normalized with `lme` methods. Finally, enhancements in dendroclimatic modeling by accounting for ecological factors are tested. We conclude that `BIOdry` package is an useful tool for studying complex ecological time-space relationships.

**1   Introduction**

Renovating Dendroclimatic Modeling in Forest Ecosystems (DMFE) is a challenging but unavoidable task in dendrochronology. Diverse methods and software for measuring Tree Rings in Wood (TRW), analyzing climate data, and compiling statistics are scattered throughout the literature (D'Arrigo et al., 2014; Matskovsky and Helama, 2016; Speer, 2010; Strokes and Smiley, 1996). Most standard approaches implement time-series decomposition at specific levels of variance (one-level modeling) to

model signs of TRW and climate fluctuations (Cook and Holmes, 1996; Grissino-Mayer, 2001). TRW data abstraction usually requires sample processing with specialized hardware (Voor Tech Consulting, 2008) and/or statistical modeling with specific software (Cook and Holmes, 1996; Grissino-Mayer, 2001; Kanatjev et al., 2014; Regent Instruments Canada Inc., 2009).

Standardized procedures for dendroclimatic analysis have been efficiently implemented for research in paleoclimatic reconstruction (Boswijk et al., 2014; Farmer and Cook, 2013; Hughes et al., 2010; Petrillo et al., 2016), forest response to water

availability (Battipaglia et al., 2009; Cocozza et al., 2014), climate change (Farmer and Cook, 2013), stable isotopes (Gerhart and McLauchlan, 2014), and other areas (Jones and Bowles, 2016; Metsaranta and Lieffers, 2009; Nehrbass-Ahles et al., 2014).



However, further research is needed to account for ecological variability in DMFE, and to integrate tree-growth-yield modeling into dendroclimatic analysis (Bowman et al., 2013; Sanogo et al., 2016; Wu et al., 2016).

Sampling dendroclimatic variables in forest ecosystems results in hierarchical sources of variability from ecological factors. For instance, variability in sample replicates of TRW from forest communities would have at least three hierarchical levels: tree-radial morphology (De Micco et al., 2014), individual-tree genetics or phenotypes (Troupin et al., 2006), and stand quality (Metsaranta and Lieffers, 2009). Additional ecological factors such as site elevation (Touchan et al., 2016), fire intensity (Jones and Bowles, 2016), tree decay (Foster et al., 2014), or water regimes (Battipaglia et al., 2009; Cocozza et al., 2014) would further complicate hierarchical variance in TRW. Meteorological records, which are used to model climatic variables in dendroclimatic analyses, also contain temporal and spatial variability (García-López and Allué, 2012; Manrique and Fernandez-Cancio, 2000).

Consequently, accounting for ecological factors in dendroclimatic variables involves statistical processing of Multilevel Ecological Data Series (MEDS), or sequences of observations ordered according to temporal/spatial hierarchies that are defined by sample designs, with sample variability confined to ecological factors (Finch et al., 2014; Pinheiro and Bates, 2000). Thorough statistical analysis of MEDS should also incorporate ecological-factor effects based on multilevel modeling or multivariate comparison. An example of multilevel modeling methods is mixed-effects regression (Bolker et al., 2009; Galecki and Burzykowski, 2013; Green and MacLeod, 2016; Oswald et al., 2012), which can be implemented to detrend dendroclimatic MEDS by considering random effects from ecological factors. An example of multivariate comparison is dissimilarity-based analysis (Goslee and Urban, 2007; Legendre, 2000; Legendre and Legendre, 2012; Oksanen, 2015), which can be used to compare and organize detrended fluctuations into common ecological-factor levels.

DMFE must also integrate tree allometry, growth, and yield methods into dendroclimatic analysis. For example, allometric scaling can help transform TRW data into other serial components of tree growth (Carrer et al., 2015; King, 2005; Peters et al., 2014; Stillwell et al., 2016; West et al., 1997). Similarly, organic growth theory has provided simplified equations (Lei and Zhang, 2004; West et al., 2001; Zeide, 1993) that can be used to subtract ontogenetic trends from TRW data. Such equations are log-linear expressions of tree growth that are easily fitted to the MEDS with the mixed-effects-modeling procedures: e.g. `lme` methods in R (Pinheiro and Bates, 2000).

Finally, DMFE could also involve time-series transformations at specific levels in the classification factors, or one-level modeling. Such modeling is important for evaluating allometric parameters for TRW (King, 2005; Stillwell et al., 2016; West et al., 1997), computing water-availability indexes from climatic MEDS (Guijarro, 2011; Manrique and Fernandez-Cancio, 2000; Martinelli, 2004; Mishra and Singh, 2010), developing statistical process control for TRW series (Bunn, 2010; Cook and Holmes, 1996; Cook and Pederson, 2011; Fritts, 1976), or for time-series smoothing and decomposition (Cowpertwait and Metcalfe, 2009; Shumway and Stoffer, 2006). One-level modeling of this kind should maintain processed MEDS structures and be adaptable to new methods in dendroclimatic modeling.

In this manuscript, we present the R-package `BIOdry`, a statistical package that integrates multilevel analysis, one-level modeling, and multivariate comparison for DMFE. Here, we explain in-package algorithms by modeling relationships in fluctuations of tree-diameter growth and drought using dendroclimatic MEDS from Iberian pine forests. We will also test the





hypothesis that considering ecological factors in MEDS improves statistical analysis for dendroclimatic modeling compared to other conventional alternatives, such as linear regression. The examples of dendroclimatic modeling developed here are also intended as instructions to guide future `BIOdry` users in implementing the package to model other ecological relationships in time and space.

## 2    Package description

R-package `BIOdry` is installed in an R session with the `install.packages()` command (R-code 1). Once installed, the package is loaded in the R environment with `require()` or `library()` commands. Current version of the package (`BIOdry V0.4`) only depends on two other R packages in CRAN: `nlme` (Pinheiro and Bates, 2000), and `ecodist` (Goslee and Urban, 2007). These are automatically loaded in R after the package has been requested. Former dependence, or `nlme`, is used to extract residuals of the processed MEDS with linear mixed effects models (detrending); while ecodist computes multivariate correlations (Mantel correlograms) between two compared MEDS. Implementation of the `BIOdry` package requires the processed MEDS to be stored as `data.frame` objects. These data structures should contain the recorded variables and time units in the initial columns, followed by the ecological factors. In the factor hierarchy, lower time units or classification factor levels are defined first, and higher levels are specified later.

The `BIOdry` package can interact with other dendrochronological libraries in R, including `dplR`, `bootRes`, and `measuRing` (Bunn, 2008; Lara et al., 2015; Zang and Biondi, 2013). For instance, the `shiftFrame()` function in the `BIOdry` package can format MEDS chronologies into `dplR` chronologies, and vice versa (R-code 1). However, to correctly format `dplR` chronologies, column names must be dot-separated labels representing the hierarchy of ecological factors, where higher ecological-factor levels are defined first and lower levels after. For example, code `'P16106.17'` is the column name of tree `'17'` in plot `'P16106'`.

Implementation of the `BIOdry` package is relatively straightforward. Although the package has diverse routines and arguments, such as the `shiftFrame()` function (Fig. 1), most DMFE procedures can be performed with `modelFrame()` and `muleMan()` functions. The following sections explain data-input requirements and function implementation.





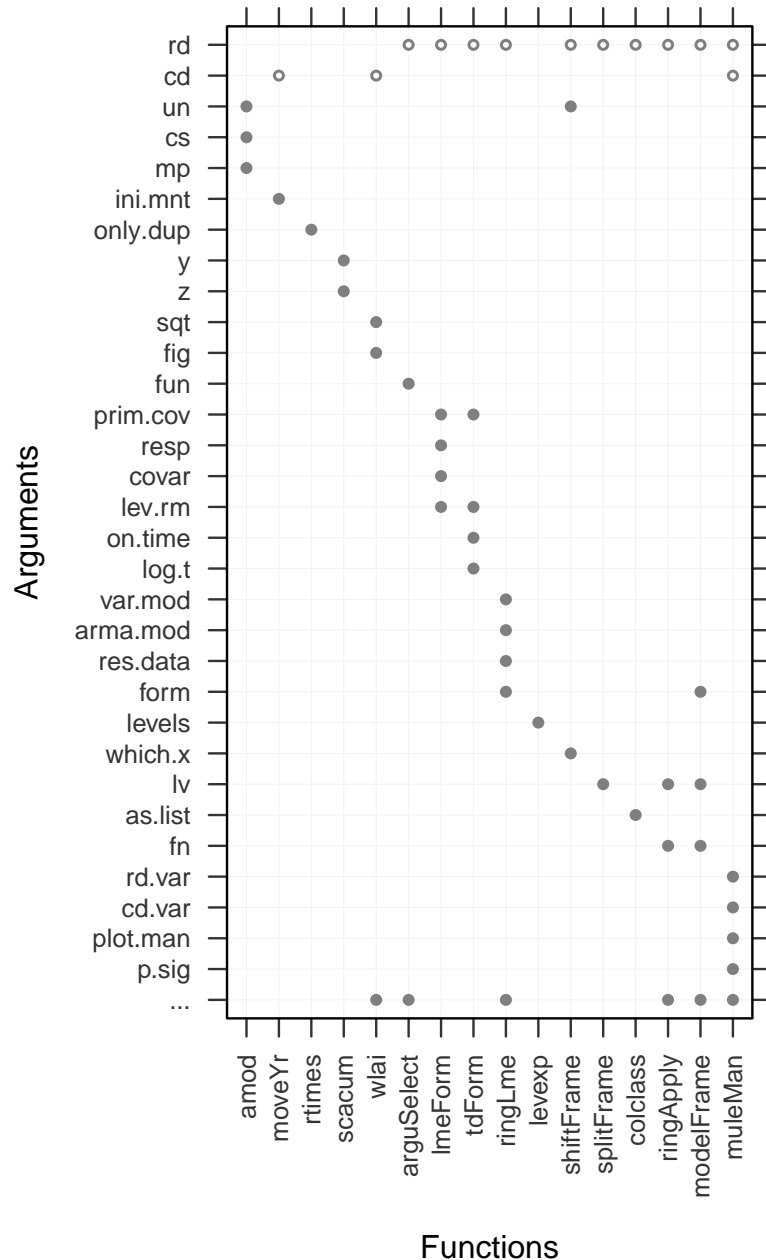

**Figure 1.** Arguments in `BIOdry` functions. Empty dots correspond to required arguments and solid dots represent arguments with defaults. The ellipsis indicates that a function can pass arguments to the previous function





## 2.1 Tree-growth-aridity relationships

Here we describe the `BIOdry` package functionalities by modeling relationships in fluctuations in tree-diameter growth and meteorological drought (aridity indexes). The fluctuations are modeled from dendroclimatic TRW and climate MEDS for pine forests (*Pinus pinaster* Ait.) in northern and east-central Spain (Table 1). These two regions have contrasting climate regimes

in which forests in central areas of the country are more affected by drought than forests in northern regions (Bogino and Bravo, 2008). We analyzed three within-stand ecological factors that are common to most dendroclimatic MEDS and represent variabilities in tree-radial morphology (core-sample replicate), tree genetics/phenotypes (tree), and stand qualities (stand). To account for source variability, two trees were selected per site, and two samples were extracted from each tree.

**Table 1.** Dendroclimatic Multilevel Ecological Data Series (MEDS) for modeling fluctuations of tree-diameter growth and drought in Iberian pine forests (*P. pinaster*). The data sets are contained in R-package `BIOdry`.

| data name | class() | Variable | Time units | Ecological factors |
|-----------|---------|----------|------------|--------------------|
| Prings05 | data.frame | TRW (mm) | year | sample, tree, stand |
| PTclim05 | data.frame | rainfall (mm), temperature (°C) | month, year | stand |
| Prings03 | vector | inside bark radii (mm) | year (2003) | within-tree codes |

## 3 Statistical formulations

The `BIOdry` package can subtract trends in dendroclimatic MEDS, using the `form` argument in `modelFrame()` (R-code 1). This argument specifies a detrending formula (Pinheiro and Bates, 2000). Currently, the `'tdForm'` or `'lmeForm'` methods can be implemented automatically with the form argument to evaluate formulas with the same names.

### 3.1 Detrending methods

The `'tdForm'` method is a linear generalization from growth theory (Zeide, 1993) which considers either time units or

classification factors in MEDS as random effects ($n/$). It has the following structure:

$$\ln(\mathbf{Y'}_{n/,t}) = y_{0,n/} + y_{1,n/}\ln(\mathbf{Y}_{n/,t}) - y_{2,n/}f(\mathbf{t}_{n/}) + \mathbf{rY'}_{n/,t}, \text{with} \tag{1}$$

$$\mathbf{rY'}_{n/,t} \sim N(0, \mathbf{R}_{n/}) \tag{2}$$

where $\mathbf{Y'}$ is relative organic growth; $\mathbf{Y}$ is cumulative organic growth; $\mathbf{t}$ is time; numerical subscripts indicate fixed effects; subscript $n/$ indicates nested random effects from classification factors: for instance, in a sample replicate in tree within stand

(stand/); $\mathbf{rY'}_{n/,t}$ is the vector of normalized residuals of $\mathbf{Y'}$; $\mathbf{R}_{n/}$ is the variance-covariance matrix of the error term that defines within-plot variability. Similarly, `form ='tdForm'` can be used, for example, to detrend inside-bark diameters.





The `'lmeForm'` method implements a more flexible linear formula, which was developed to enable users to formulate their own multilevel linear expressions. For instance, `form = 'lmeForm'` can be implemented to detrend normalized aridity indexes.

Monthly average temperatures (°C) and monthly precipitation sums (mm) in `PTclim05` can be transformed into annual aridity indexes from Walter-Lieth diagrams (Manrique and Fernandez-Cancio, 2000). In other words, algorithms in the `BIOdry` package can construct such diagrams and compute the aridity indexes as ratios between dry and wet areas in the diagrams. This method integrates seasonal precipitations and temperatures to identify periods of relative water surplus and deficit (Kempes et al., 2008). Square root transformations of the aridity indexes are poorly autocorrelated (Lara et al., 2013), thereby avoiding the frequent and problematic need for further modeling of autocorrelations in residual structures of the climatic variables in dendroclimatology (Cook and Pederson, 2011). For example, the `'lmeForm'` method formula can have the following structure:

$$\text{sqt}(\mathbf{AAI}_{m/,t}) = \alpha_{0,m/} + \alpha_{1,m/}(\mathbf{t}_{m/}) + \mathbf{rAAI}_{m/,t}, \text{with} \tag{3}$$

$$\mathbf{rAAI}_{m/,t} \sim N(0, \mathbf{R}_{m/}) \tag{4}$$

where sqt is square root transformation; $\mathbf{AAI}$ is a vector of annual aridity indexes; $\mathbf{t}$ is the time, $m/$ indicates the nested random effects defined by the classification factors in the climatic MEDS. In the forest stand, for example, note that for comparison purposes the climatic data should have at least one classification factor in common with ecological-factor levels of the dendrochronological MEDS. As an example, in the location of the studied forest, $\mathbf{rAAI}_{m/,t}$ is the vector of normalized $\mathbf{AAI}$ residuals; $\mathbf{R}_{m/}$ is the variance-covariance matrix of the error term that defines within-ecological-factor variability.

### 3.2 Variance functions and correlation structures

Heteroscedasticity and serial autocorrelation of the detrended multilevel series $\mathbf{rY'}_{n/,t}$ and $\mathbf{rAAI}_{m/,t}$ can be modeled with arguments in `lme` methods that are specified in `modelFrame()`: e.g. weights and correlation arguments in `nlme` package (Pinheiro and Bates, 2000). For instance, a variance model for the lowest level in the detrended growth series is:

$$\text{Var}(\mathbf{rY'}_{c,t}) = \sigma^2 \left( \varrho_1 + |v_{c,t}|^{\varrho_2} \right)^2 \tag{5}$$

where $\sigma^2$ is the variance, $\varrho_1$ and $\varrho_2$ are constants, $v_{ct}$ had the same fixed-effects structure, and $\varepsilon_{ct}$ are the modeled residuals at the lowest level in $n/$ (Eqs. 1 and 3). Autocorrelation can be assessed with the empirical autocorrelation function. For example, the autocorrelation model for the lowest level of inside-bark diameters can be

$$\hat{\rho}(l) = \frac{\sum_{c=1}^{n_i} \sum_{t=1}^{n_c-l} r_{c,t} r_{c,(t-l)} / N(l)}{\sum_{c=1}^{n_i} \sum_{t=1}^{n_c} r_{c,t}^2 / N(0)} \tag{6}$$

where $r_{c,t}$ are standardized residuals, $l$ is the lag in years, and $N(l)$ is the number of residual pairs defining the numerator of $\hat{\rho}(l)$. When present, autocorrelation could be modeled with an autoregressive-moving average model `ARMA (1,1)`, which



has an exponentially-decaying auto-correlation function for lags $\geq 2$. For the lowest level in any of the subtracted residuals, the autocorrelation model is

$$\varepsilon_{c,t} = \phi \cdot \varepsilon_{c,(t-1)} + \theta \cdot a_{c,(t-1)} + a_{c,t} \tag{7}$$

where $\phi$ was the autoregressive parameter; $\theta$ was the moving average parameter; and $a_{c,(t-1)}$, and $a_{c,t}$ were the noise terms.

## 3.3 Multivariate comparison

Mantel correlograms between two MEDS with a common classification factor are established by comparing distances in one of the MEDS with sets of binary model matrices that specify membership in the other MEDS classes. Euclidean distances from both MEDS are standardized to z-scores. The Mantel statistic is

$$r(d) = \frac{\sum_i^n \sum_j^n w_{ij} z_{ij}}{\sum_i^n \sum_j^n w_{ij}} \tag{8}$$

where $d$ is the distance class from one of the MEDS; $z_{ij}$ is the distance between each pair $i$ and $j$ from the other MEDS; $w_{ij}$ is a weight for the pair: typically 1 if $z_{ij}$ was in $d$ and 0 if it was not. Number of classes $d$ is calculated using the Sturges rule (Legendre and Legendre, 2012).

## 4 Package implementation

### 4.1 Inputs: MEDS

We have formatted two dendroclimatic MEDS into `Prings05` and `PTclim05` data sets, which are included in the package (Table 1). These can be loaded into R with the `data()` function (R-code 1).





```
## Install and load BIOdry package:
> install.packages('BIOdry')
> require('BIOdry')
## loading Dendroclimatic MEDS:
> data(Prings05); data(Pradii03); data(PTclim05)
## data formatting:
> dpChron <- shiftFrame(Prings05)  # dplR chronologies
> Prings05 <- shiftFrame(dpChron) # MEDS chronologies
## modeling tree diameters:
### lists of arguments:
> fn_td <- list('rtimes','scacum','amod')
> lv_td <- list('tree','sample','sample')
> MoreArgs_td <- list( z = 2003,
                      mp = c(1,1),
                      un = c('mm','cm'))
### recursive modeling:
> diameters <- modelFrame(rd = Prings05,
                          fn = fn_td,
                          lv = lv_td,
                     MoreArgs = MoreArgs_td,
                          y = Pradii03,
                        form = 'tdForm',
                       log.t = TRUE,
                      method = 'REML')
## drought modeling:
### List arguments
> fn_d <- list('moveYr','wlai')
> lv_d <- list('year','year')
> MoreArgs_d <- list(ini.mnt = 'Oct',
                         sqt = TRUE)
### recursive modeling:
> drought <- modelFrame(rd = PTclim05,
                        fn = fn_d,
                        lv = lv_d,
                   MoreArgs = MoreArgs_d,
                     form = 'lmeForm') # or form = NULL
```

**R-code 1.** Recursive modeling of tree-diameter growth and drought fluctuations with `BIOdry` package.





The *P. pinaster* TRW (`Prings05` data set, Table 1) belongs to a more extensive TRW chronology which was processed in previous studies (Bogino and Bravo, 2008) by measuring tree-ring widths in polished-core samples (5 mm diameter) using the Windendro program (Regent Instruments Canada Inc., 2009) and cross-dating the tree-ring chronologies using COFECHA software (Grissino-Mayer, 2001). Time units for the TRW are formation years spanning from 1810 to 2005. The cross-dating records suggested that maritime pine chronologies had high SNR: ($23-28$ in northern Spain, and $38-61$ in east-central Spain) and high EPS ($0.96$ in northern Spain, and $0.98$ in east-central Spain).

Climatic records (`PTclim05` data set, Table 1) were provided by the State Meteorological Agency of Spain (AEMET). Time units in this data set are recorded months from January to December, and observed years spanning from 1951 to 2005. The classification factor corresponds to location labels of closer dendrochronological plots, which define one sample level (`plot`).

One reference radii vector (`Pradii03`, Table 1) is also provided in the package to give directions about vector formulations in recursive evaluation of functions (Section 4.2, R-code 1). The numeric vector contains inside-bark radii that were measured in 2003. Allometric parameters from a previous study (Lizarralde, 2008) were applied to over-bark diameters at breast height to estimate inside-bark diameters. The vector names are within-tree codes. Until recently, core samples often lacked the initial rings of the pits producing truncated growths; so these reference radii can be used to scale cumulative growth.

## 4.2 Derived inputs: one-level modeling and recursive formulation

Modeling dendroclimatic fluctuations implies diverse formulations in one-level modeling (see examples in Table 2), but multiple implementation of such functions can be burdensome. Time-series synchronization of dendroclimatic MEDS enhances convergence of mixed-effects parameters (Pinheiro and Bates, 2000) and avoids important biases in the extracted fluctuations (Bowman et al., 2013).




**Table 2.** Five one-level functions used to derive cumulative tree diameters (cm) and aridity indexes (dimensionless) from dendroclimatic MEDS, using R-package `BIOdry`. Required arguments (Arg.) for evaluation of the functions are specified.

| Name | Details | Arg. | Arg. definition |
|---|---|---|---|
| rtimes | Unique observations in time-units data with replicates (time-series replicates) are excluded to avoid biases during subsequent multilevel detrending Bowman et al. (2013). | only.dup | logical: (TRUE, FALSE). Extract only relative times that are duplicated. If TRUE then unique observations are replaced with NA. If all computed times are unique then this argument is ignored. |
| scacum | Cumulative sums of time-series replicates (e.g. radial increments) are scaled on reference constants (e.g. individual tree diameters). | y | numeric constant, or vector if the processed time-series replicates have several levels, to scale the computed cumulative values. If NA then the computed cumulative sums are not scaled. |
| | | z | NA, numeric constant, or vector if the processed time-series replicates have several levels. Reference time(s) in range(s) of the vector names to scale the cumulative values. If NA then maximum value in the range is used. |
| amod | Simple allometric model: $y = a(2x)^b$ is recursively evaluated to derive allometric components of organisms from longitudinal variables (e.g. Cumulative radial increments). | mp | numeric vector with allometric parameters: $a, b$. default c(0.5,1) maintains the original radii, c(1,1) produces diameters, and c(0.25 * pi, 2) computes basal areas. This argument can have more than two parameters: c(a1,b1,a2,b2, ..., an,bn), with n being the number of times that an allometric model will be recursively implemented. |
| | | un | NULL, or bidimensional character vector with the form c(initial, final) to transform SI units of the processed variable. The SI units can be expressed in micrometers 'mmm', millimeters 'mm', centimeters 'cm', decimeters 'dm', or meters 'm'. If NULL then original units are maintained. |
| moveYr | Monthly records in time-series replicates (usually of climate) are labeled for the years can begin in a month other than January. | ini.mnt | character, or numeric from 1 to 12. Initial month of the seasonal year. If character then the months are built-in constants in R-package base. Default 'Oct' makes the years begin in October, for example. |
| wlai | Annual aridity indexes from Walter-Lieth diagrams are computed from monthly precipitation sums and monthly average temperatures. | sqt | logical. Print the square root of the aridity index. If TRUE then computed aridity index is normalized with a square root transformation. |

TRW synchronization implies several within-level formulations of one-level functions, such as within-sample transformation of time units (usually formation years) into time indexes ($t = 1, 2, \cdots, n$) and within-tree synchronization of time indexes into common time-series windows (Table 2). Additional one-level modeling is required to compute tree diameters: for example, within-sample TRW should be cumulated, scaled around the reference radii, and multiplied by two. Here, we computed the

5     diameters via allometric scaling (see the `amod` function in Table 2).

The recursive `modelFrame()` function in the `BIOdry` package simultaneously evaluates diverse one-level functions (R-code 1). As a wrapper for the higher-order `Map()` function (default R-package base), `modelFrame()` recursively applies individual one-level functions at specific levels in the MEDS while preserving the initial factor-level structure in the derived inputs. One-level function arguments are passed to `modelFrame()` using the formulation suggested in the base package

10     for `Map()` and `mapply()` functions: one-level function parameters are specified in either a `MoreArgs` list argument or in the level parameters vectors, depending on whether they are constants or vectors, respectively (R-code 1).





The diverse `modelFrame()` funcionalities are controlled with at least six kinds of parameters: one dendroclimatic MEDS (`data.frame` object) in the `rd` argument; three list arguments (`lv`, `fn`, and `MoreArgs`); level parameter vectors (`y`, `z`, etc.); and an alternative detrending method (`form`). The list arguments make `modelFrame()` process data in `rd` by recursive evaluation of functions specified in `fn` at the levels defined in `lv`. An interesting feature of `modelFrame()` is its capacity for

expanding incomplete level parameter vectors for all the ecological-factor levels in the processed MEDS. To accomplish this, such vectors should contain observations in some of the studied levels and be labeled with dot-separated names representing the hierarchy of ecological factors. This is similar to what was explained for `dplR` chronologies in Section (2).

### 4.2.1 Tree-diameter modeling

To compute inside-bark diameters, three in-package functions for one-level modeling are recursively implemented here:

`rtimes()`, `scacum()`, and `amod()` (Table 2), which develop TRW synchronization, compute cumulative TRW sums, and implement allometric scaling, respectively. The `amod()` function is recursive: diverse pairs of allometric parameters can be formulated at the same time to derive components in tree-growth morphology, such as tree diameters, basal areas, above-ground biomasses, etc.

For example, one pair of parameters `mp = c(1,1)` was implemented to model tree diameters (see `diameters` object

in R-code 1). The data processed in `rd` was the `Prings05` object. The three one-level functions were formulated in the `fn` argument and the levels of evaluation were specified with the `lv` parameter. The `MoreArgs` list in the diameters data set contained the arguments that were constant for the evaluated levels: year of measurement of the reference radii in `z`, allometric parameters in `mp`, and transformation of SI units for allometric modeling in `un` (Table 2). The `y` parameter contained the reference radii vector at tree level.

### 4.2.2 Drought modeling

Formulating parameters for computing aridity indexes from `PTclim05` in `modelFrame` is similar to what was explained for modeling tree diameters (see `droughts` object in R-code 1). Here, new `moveYr` and `wlai` functions were specified in the `fn` argument (Table 2); the `moveYr` function transforms time units into climatic variables for the years and can begin with October (`ini.mnt = 'Oct'`, beginning of tree-growth season in Spain). The `wlai` function calculates normalized annual aridity

indexes (`sqt = TRUE`, square root transformation) from the transformed climatic records. Both functions were evaluated at the year level with the `lv` argument.

### 4.3 Dendroclimatic detrending

The `modelFrame()` function subtracts trends in dendroclimatic MEDS when the `form` argument specifies any of the in-package `'tdForm'` or `'lmeForm'` methods. The function fits mixed-effects models and extracts multilevel residuals/fluctu-

ations. The `'tdForm'` method produces in-package algorithms to fit Eq. (1) to subtract the ontogenetic component from the TRW. The second `'lmeForm'` method enables users to modify the linear structure of the detrending method; see the example



in Eq. (3). Users can also define their own `lme` formulas according to these two formulated examples using arguments in `tdForm` and `lmeForm` functions, as in Fig. (1). Arguments in the `form` method or in the `lme` function are also specified into `modelFrame()`, see for example `log.t = TRUE` and `method = REML` arguments in `tdForm` and `lme` functions in R-code 1, respectively.

After the detrending process is developed, the `modelFrame()` function produces a list with three elements: the model parameters (`model`), the extracted residuals of the model (`resid`), and a call function for further parameter updating (`call`). These are accessed by using either the dollar sign (R-code 2) or bracket operators: e.g. `diameters[['resid']]`.

## 4.4   Model updating and serial normalization

Another objective of the package is to assist with the normalization of MEDS fluctuations. The `lme` models in `modelFrame()`
objects can be inspected with regular `lme` methods such as empiric autocorrelation function (ACF), anova (`lme.anova`), model summaries (`lme.summary`), and others (Pinheiro et al., 2016; Pinheiro and Bates, 2000). These methods are implemented to develop serial normalization in the extracted residuals and to change previously formulated arguments.

For example, within-stand residual auto-correlation in the diameter list was inspected with the ACF (R-code 2). The inspection process suggested that residuals in the diameters list be autocorrelated at three lags. Consequently, the diameter model
required additional `lme` modeling to remove the three-lag autocorrelation. This was carried out by updating `modelFrame()` with new `lme` arguments, as seen in the update function (R-code 2). The correlation parameter is used to model the within-group residual autocorrelation. This time, an ARMA model for lags $\geq 2$ was fitted (`correlation = corARMA(p = 1, q = 1)`). After developing the updating process, the ACF function indicated that new object `diameters_ac$'resid'` was uncorrelated.

The `update()` function is also useful for modifying arguments in `modelframe()` objects. In this case, we specified `lev.rm = 'plot'` (`tdForm` method) to remove the higher plot level from the mixed model in the updated `diameters_ac` list (R-code 2). Changing levels in this way facilitates comparison of models with different random effects structures (Table 3).



```
## Models inspection:
> summary(diameters$'resid')
> summary(drought$'resid')

## Residual-autocorrelation inspection:
> plot(ACF(diameters$'model', maxlag = 10),
        alpha = 0.01) # the diameter residuals are
                        # autocorrelated at three lags!

> plot(ACF(drought$'model', maxlag = 10),
        alpha = 0.01) # the aridity-index residuals are not
                        # autocorrelated

## Updating the 'modelFrame' function:
## Random effect on 'plot' is subtracted from
## 'diameters' model, and within-group residual
## autocorrelation is modeled:
> diameters_ac <- update(diameters,
                    lev.rm = 'plot',
                  correlation = corARMA(p = 1, q = 1))
> summary(diameters_ac$'model')

## Second inspection of within-group residual autocorrelation:
> plot(ACF(diameters_ac$'model',maxlag = 10,
            resType = 'n'),
              alpha = 0.01) # within group residuals are not
                              # autocorrelated now

##Multivariate comparison:
> interactions <- muleMan(diameters_ac,
                      drought,
                      nperm = 10^3)
```

**R-code 2.** Multilevel normalization and multivariate comparison with `BIOdry` package.



## 4.5 Multivariate comparison

The `BIOdry` package can correlate `MEDS` that have common time units and classification factors with Mantel correlograms (see section 3.3). Multivariate comparison is developed with the `muleMan()` function in the package.

For example, `muleMan()` was implemented to compare Residuals of Annual Diameter Increment (RADI) and Residuals of Annual Aridity Index (RAAI) fluctuations that were stored in `diameters_ac` and `drought` objects, respectively (R-code 2). The multivariate comparison suggested that relationships between detrended tree diameters and detrended aridity indexes depended on within-plot factors, and plot location (sites) was the most important driver of the modeled dendroclimatic relationships (Fig. 2). Within-plot variability affected the patterns of the computed correlations over time, while within-tree variability influenced correlation trends and scales across the computed distance classes. Trends in the two sites presented oscillating pulses, and pulse frequency was significant for distance classes greater than zero (usually around distance classes 2 or 6). Dendroclimatic interactions were stronger in east-central Spain (P16106) than in northern Spain (P44005). The Mantel correlogram also indicated that dendroclimatic relationships were affected by within-sample variabilities, such as radial-increment morphologies in this case. Sample replicates of the same trees indicated both significant and no significant responses to drought (Fig. 2, dark and grey circles), and discrepancies were more evident in P16106 than in P44005.

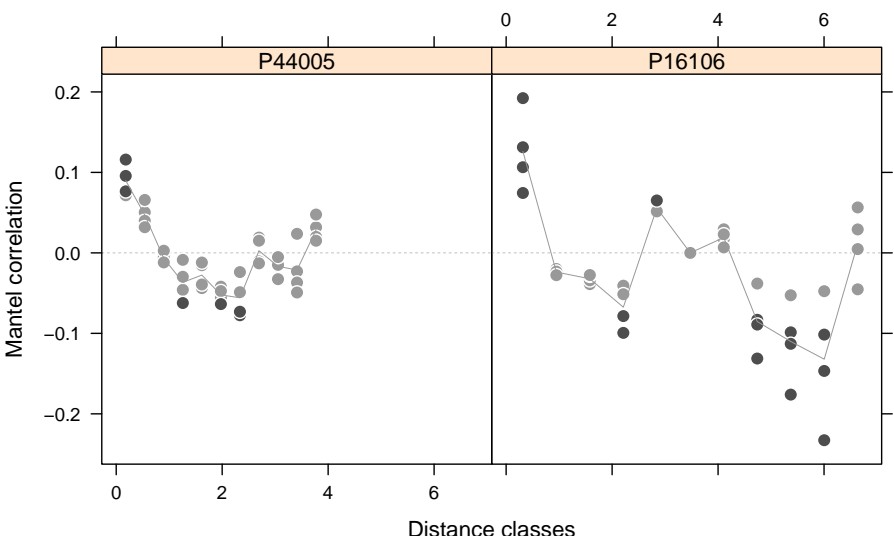

**Figure 2.** Multivariate correlations between normalized Residuals of Annual Diameter Increment (RADI) and normalized Residuals of Annual Aridity Index (RAAI) at nested levels within two stands: two cores per tree, and two trees per stand. Darker circles indicate significant correlations ($p \geq 0.05$).





## 4.6 Outputs

Both `modelFrame()` and `muleMan()` objects maintain the time units and ecological-factor structures in processed MEDS, which can be visualized in R with trellis-graphics packages such as `lattice` or `ggplot2`. Evaluation of `modelFrame()` produces `lme` models with mixed-effects parameters and `data.frame` objects of derived inputs, depending on whether the

data is detrended or not. After developing the detrending processes, output lists from such a function would contain model parameters (`model`), function calls (`call`), and extracted residuals of the model (`resid`). Implementation of the `muleMan()` function produces `data.frame` MEDS with Mantel correlations.

## 5   Trend-subtraction testing

This study also aims to validate multilevel-detrending procedure in the `BIOdry` package. Consequently, we tested methods in

`BIOdry` to detrend MEDS variables by accounting for residual variability in ecological factors (Table 3).

**Table 3.** log-Likelihood radio tests (logLik) comparing parameters fitted on `tdForm` and `lmeForm` formulas for detrending dendroclimatic MEDS of Annual Diameter Increment (ADI) (cm year$^{-1}$) an Annual Aridity Index (AAI) (dimensionless), respectively. Models with only fixed effects (fe) are compared with models with random effects which are progressively expanded by considering classification factors on sample replicate (sample), sample within tree (tree/), and tree/ within plot (plot/), and by accounting for residual heteroscedasticity (h) and residual autocorrlelation (a).

| Model | Structure | Nr. | df | BIC | logLik | Test | L.Ratio | p.val |
|-------|-----------|-----|-----|---------|---------|--------|---------|-------|
| `tdForm` | fe, plot/, a | 1 | 15 | 1344.33 | -620.47 | | | |
| fitted on | fe, plot, h | 2 | 15 | 1453.76 | -675.18 | | | |
| ADI | fe, plot/, h, a | 3 | 17 | 1333.87 | -608.35 | 2 vs 3 | 133.67 | 0.00 |
| | fe, plot/ | 4 | 13 | 1487.93 | -699.16 | 3 vs 4 | 181.63 | 0.00 |
| | fe, tree/ | 5 | 10 | 1469.02 | -700.05 | 4 vs 5 | 1.76 | 0.62 |
| | fe, sample | 6 | 7 | 1566.10 | -758.92 | 5 vs 6 | 117.76 | 0.00 |
| | fe | 7 | 4 | 1545.42 | -758.92 | 6 vs 7 | 0.00 | 1.00 |
| `lmeForm` | fe, plot, a | 1 | 7 | 60.62 | -14.26 | | | |
| fitted on | fe, plot | 2 | 5 | 57.40 | -17.24 | 1 vs 2 | 5.96 | 0.05 |
| AAI | fe | 3 | 3 | 50.19 | -18.22 | 2 vs 3 | 1.95 | 0.38 |

   The `modelFrame()` function was implemented to compute MEDS of tree-diameter growth and aridity indexes from `Prings05` and `PTclim05` data sets, respectively (R-code 1). Parameters in `tdForm` and `lmeForm` equations were fitted on the modeled MEDS both with `lm` function (R-package `base`) as well as with `modelFrame()` function (`BIOdry` package). The `lm` fittings did not consider residual variability from ecological factors, and were used to test whether accounting for

ecological factors and within-factor residual normalization improved the lme models. The `modelFrame()` function was also




implemented to extract model parameters with different random-effect structures and to develop the residual normalization; see examples in the R-code (2) to update `modelFrame()` function with new structures in random-effect ecological factors and for autocorrelation modeling.

Parameters between fitted models were compared with Bayesian Information Criterion (BIC) and log-Likelihood radio tests
(logLik) in the `nlme` package (Pinheiro et al., 2016). Parameter significances in `lme` summaries of selected `tdForm` or `lmeForm` models were also used to test the theoretical bases of the detrending methods in the `BIOdry` package.

### 5.1 Tree-diameter fluctuations

BICs and logLiks indicated that `tdForm` mixed-effects models with random-effects variability that was nested within tree/ or within plot/ fitted better than the `tdForm` model with only fixed effects (Table 3, upper side). Extending the mixed-effects
models with corrections for heteroscedasticity and autocorrelation further improved BICs and logLiks parameters. However, modeling heteroscedasticity produced non-significant parameters in the fixed effects of the extended `tdForm` mixed-effects model (p ≤ 0.28). Consequently, heteroscedastic modeling was avoided, and the best `tdForm` model included random effects nested within plot/ as well as a correction for autocorrelation only (BIC = 1344.33).

**Table 4.** Fixed-effect coefficients (fe) and random-effect standard deviations (sd) of best fitted models to detrend Annual Diameter Increment (ADI) (cm year$^{-1}$) and Annual Aridity Index (AAI) (dimensionless). *int.* are model intersects; **D** are inside-bark diameters (cm); and **t** are recorded times (time indexes, years). Responses are transformed with square root (ln) and natural logarithm (sqt) operators.

| Method | Response | Parameter | fe | sd | | |
| | | | Value (p.val) | plot | plot/tree | plot/tree/sample |
|---|---|---|---|---|---|---|
| `tdForm` | ln(ADI) | *int.* | −2.47 (0.03) | 0.40 | 0.85 | 0.03 |
| | | ln(**D**) | 1.27 (0.03) | 0.00 | 0.00 | 0.02 |
| | | ln(**t**) | −0.87 (0.00) | 0.08 | 0.21 | 0.00 |
| `lmeForm` | sqt(AAI) | *int.* | -18.93 (0.00) | 0.07 | | |
| | | **t** | $9.98 \times 10^{-3}$ (0.00) | 0.00 | | |

P-values below 0.033 in fixed-effect parameters of the selected `tdForm` mixed-effects model (Table 4) suggested that tree-
diameter growth derived from TRW conformed to the theoretical growth patterns implied by the theoretical growth equation during the last 50 years. Detrended within-plot diameters fluctuated around zero over time and exhibited constant variance (Fig. 3).





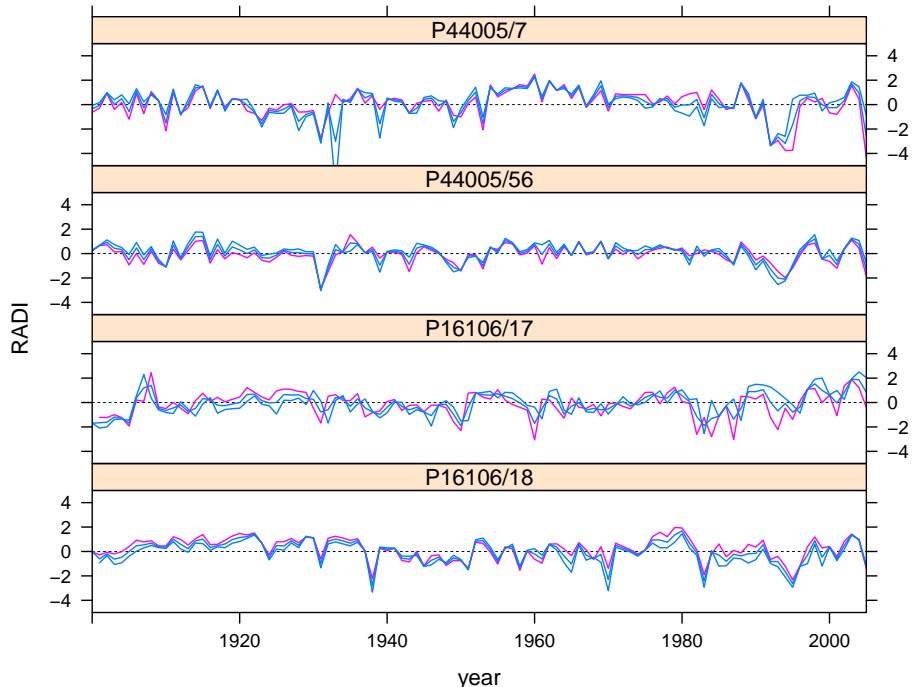

**Figure 3.** Normalized Residuals of Annual Diameter Increment (RADI). Central lines indicate average patterns in fluctuations. Extreme lines indicate patterns in fluctuations of sample replicates. Fluctuations are plotted in panels nested within tree/ to better illustrate sample variability. However, the residuals were extracted with a `tdForm` mixed model with random effects nested within plot/ and a correction for residual autocorrelation.

## 5.2 Aridity-index fluctuations

Parameters for the mixed-effects `lmeForm` model with one random effect per plot exhibited non-significant differences with regard to the corresponding fixed-effects model (Table 3, lower side). However, this random effect was maintained to test significance in within-plot residual autocorrelation and heteroscedasticity using `lme` methods. Inspection of scatter plots of within-plot residuals suggested no evidence of heteroscedasticity, and expanding the mixed model with variance-parameter structures did not enhance the model (singular gradients). Similarly, the empirical autocorrelation function for the within-plot residuals indicated no significant within-plot serial autocorrelations ($p \leq 0.01$). Consequently, the linear `lmeForm` mixed model was not expanded. Significant fixed effects in the mixed-effects models ($p \leq 0.00$) suggested that drought had increased throughout the study area during the last 50 years, though the slope was very low. Detrended within-plot aridity indexes fluctuated around zero over time and displayed constant variance (Fig. 4).





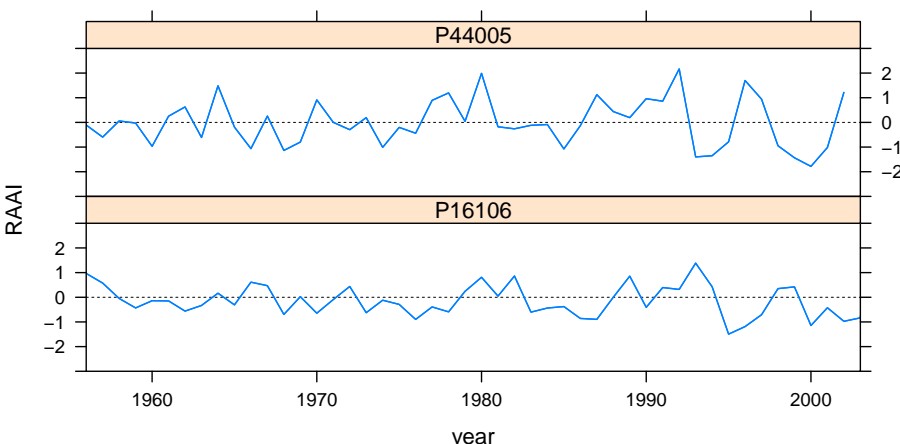

**Figure 4.** Normalized Residuals of Annual Aridity Index (RAAI). Residuals were extracted with a `lmeForm` mixed model with one random effect per plot.

# 6 Discussion

## 6.1 `BIOdry` **package**

The `BIOdry` package is a useful tool for DMFE that integrates recursive evaluation of one-level functions, multilevel detrending, and multivariate comparison of MEDS. Package users will find it relatively easy to use, as it only requires implementation
of the `modelFrame()` and `muleMan()` functions. Most arguments in other functions of the same package, or in the `nlme` library, can be implemented with the same `modelFrame()` function.

The most important contribution of the `BIOdry` package is its capacity to enhance DMFE by incorporating ecological factors in MEDS. The package has features to statistically test detrending modeling by nesting ecological factors. For example, progressive expansion of the mixed-effects ADI model suggested that accounting for within-plot variabilities and residual
autocorrelation improved model parameters (Table 3) and produced normalized fluctuations of tree-diameter growth (Fig. 3). Likewise, expansion of the mixed AAI model also produced significant fixed-effect parameters and normalized drought fluctuations (Fig. 4), though statistical improvement of the mixed AAI model by nesting within-location variability was not conclusive.

Tree-diameter modeling suggested that statistical methods in the package effectively controlled other random processes
that might impose physical and biological effects on tree-diameter growth, but which were not controlled in this study. These include local variations in soil nutrient and soil water availability, tree provenance, forest management, and other ecological factors. Aridity index modeling suggested that these methods helped to identify the need for further research on climatic factors affecting AAI patterns: specifically, additional meteorological records indicating local conditions in studied stands. Future studies could incorporate these and other factors into similar mixed-effects models and quantify their explicit influence
on dendroclimatic interactions.



Multivariate comparison facilitated the integration of regional-climatic analysis with stand-level data processing, thus eliminating the need for averaging dynamic time series and ensuring the removal of spurious relationships. Permutation tests in Mantel-correlogram computations guaranteed that significant correlations were not simply the consequence of comparing successive anomalies over time.

Recursion in the package simplified modeling in tree-growth-drought relationships. Recursive formulation efficiently processed dendroclimatic variables at specific levels in the ecological factors while maintaining the column structures of processed MEDS. With other modeling approaches, deriving variables from MEDS can require a great deal of time and computational resources. For instance, reordering datasets after variables have been transformed usually requires additional data processing with other R-libraries or office-automation software (Wickham et al., 2011).

The recursive formulation applied to dendroclimatic data enhanced statistical computation by properly scaling the data in space and time (Bowman et al., 2013). For example, three one-level functions were simultaneously evaluated from TRW data: within-tree time indexing, within-sample radial scaling, and within-sample diameter computing. Likewise, two one-level functions were recursively formulated for the meteorological data for within-year seasonal scaling and within-year AAI computing. Though these dendroclimatic MEDS were dissimilar, the corresponding recursive formulations required only individual

`modelFrame()` implementations. Such spatial-temporal scaling in the processed MEDS also helped improve convergence of detrending parameters, because within-group observations were synchronized: i.e. the TRW were contained in common time-series windows and had common time indexes.

Another clear advantage of the `BIOdry` package is its capacity for allometric modeling of tree-growth components using the `amod()` function. Modeling fluctuations in tree-growth allometry, rather than simple TRW increments, has become essential

in dendrochronology (Carrer et al., 2015) and helped us understand the effects of water availability on tree morphology (King, 2005). In the examples, we computed tree diameters by modeling the maritime pine dataset (`Prings05`) with the `modelFrame()` function and the argument `mp = c(1, 1)` in the `amod()` function. However, other tree-growth variables can also be computed by changing the `mp` argument. For example, tree basal areas (`mp = c(0.25 * pi, 2)`) or other tree-biomass components with recursive implementation of `mp` parameter pairs could be studied.

The `BIOdry` package also allows the user to subtract trends in organic variables in MEDS using the `'tdForm'` method. The Equation in this method has biological meaning (Lei and Zhang, 2004; Zeide, 1993). We implemented such an equation in a mixed-effects fitting using the `modelFrame()` function to detrend tree-diameter growth. This fitting illustrated that tree-diameter growth conformed to the growth theory implied by such an equation. It also suggested that a log-linear expression in the `'tdForm'` method can easily be fitted with mixed-effects regression as well as with other conventional regression

methods, such as `lm` fitting. Consequently, the `BIOdry` package could be implemented with other ecological research subjects to model ontogenetic growth of organic variables in MEDS.

Despite the clear statistical enhancements in DMFE using the multilevel approach, it does not meet the requirements for properly replicated MEDS in all cases. For instance, the low degrees of freedom in the mixed-effects AAI model could not account for the random effect on plot. This within-plot climatic uncertainty was not important because fixed-effects parameters




in the mixed-effects model were significant and within-plot fluctuations were centered around zero (Fig. 3). More extensive meteorological records would be needed to test the importance of this location effect more appropriately.

## 6.2 Tree-growth-aridity relationships

Modeled dendroclimatic relationships agreed with previous conclusions that drought conditions in central and southern Spain have affected tree-diameter growth with regard to the tree-growth-drought relations observed in northern areas of the country (Ruiz-Sinoga and Martínez-Murillo, 2009; Sabaté et al., 2002). Again, the most important factors affecting the relationships between RADI and RAAI were the time span of the analysis and latitudinal location of the stands. In east-central Spain, RADI have responded to RAAI during the last 50 years, and the magnitude of RAAI responses has progressively increased. Increasing drought during the last 50 years may have altered annual balances of soil moisture, or nutrients in litter and soil organic matter relative to levels in the recent past (Ogaya et al., 2003; Sánchez-Gómez et al., 2011). Consequently, ecosystem productivity may have shifted to new states of equilibrium (van der Molen et al., 2011). Trees in the northern Spain were less sensitive to annual droughts because drought duration may not have altered long-term soil stores, thereby preserving a buffer against adverse effects on trees. Such patterns resemble dendroclimatic relationships established in the modeling of tree-biomass growth and aridity fluctuations in a previous study (Lara et al., 2013).

## 6.3 Future implementations in plantation forestry

Implementing the functionalities in the `BIOdry` package to study the effects of ecological factors on forest-ecosystem dynamics could have great relevance for managing natural forests or monocultures, or for understanding the role of species mixtures in plantation forestry. In the latter case, effects from ecological factors representing complementary resource uses (root stratification, nutrients, facilitative improvements, etc.) or even commercial productivity might be modeled with the package. We hope that other scientists can implement the `BIOdry` package to study the effect of such factors on many species and at different successional stages.

## 7 Conclusions

the `BIOdry` package integrates one-level modeling, multilevel analysis, and multivariate comparison of `MEDS`. The package routines facilitate rapid handling of classification factors to derive new inputs, while preserving the structure of the classification factors. Derived inputs can be detrended, normalized, or compared through implementation of the `modelFrame()` and `muleMan()` functions. These features make `BIOdry` package a valuable tool for studying complex ecological time-space relationships.

## 8 Code availability

The in-package code is available on https://cran.r-project.org/web/packages/BIOdry/index.html




## 9 Data availability

Data sets processed here are contained in the `BIOdry` package

*Acknowledgements.* This research has been supported through funds from the Colombian government (Colciencias) within the framework of the National Program for Doctoral Studies Abroad (Year 2012, Number 568). This study was also financed by the FORMIXIG project
5 (AGL2014-51964-C2-1-R) funded by the Spanish Ministry of Economy, Industry and Competitiveness. Our thanks to the researchers at the UVa-INIA Sustainable Forest Management Research Institute for their assistance in core sampling and for their ideas related to the development of the algorithm; in particular Cristobal Ordoñez, Ana de Luca, José Riofrio, and Cristina Prieto. We would also like to thank the anonymous reviewers for their suggestions and comments.



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
