# Peer review of "R-Package BIOdry: DendroClimatic Modeling from Multilevel Ecological Data Series"

_Biogeosciences, 2016_

## Referee Comment (RC1) · Anonymous Referee #2 · 31 Jan 2017

The present study introduces new R-package which can be useful in tree-ring research, but also in other time-series analyses. The manuscript is clearly written and contains enough detail to follow the analytical protocol employed. Package is already available at CRAN. However, I feel that such type of manuscript should be mostly dedicated to the description of individual functions and how user can modify this and what will be the output. However, this paper is far from clear cookbook for potential package user, which I believe is really pity. There is a lot of space dedicated to explanation of mathematical processes, but this should be rather in the Appendix (honestly, most of the readers and potential users will not care about it, it is your responsibility that used methods are correct, with this description you will not increase the understanding for people which simply do not understand these methods, just to want to use them as a tool without deep knowledge how it works). I also found some errors in the script,

which is really something what I would not expect from the manuscript describing the package. There are some Figures in the manuscript, which I suppose are produced by some function of the package. However, this is really not clear and it is only my opinion. I would expect that there will be clear line of script shown in manuscript, as basically all packages published in dendrochronology did, and this line will be accompanied by short description what it is doing and if there is a figure produced, then it should be explicitly written. Authors mention that "the BIOdry package can interact with other dendrochronological libraries in R, including dplR, bootRes, and measuRing", however, they show only one example of communication of dplR package and some connection with other packages is not shown at all. In addition, even this one example is actually quite rather showing that these packages are not able to communicate, as the format of data is different and I do not understand why it is not unified or there is not some function for transformation to commonly used and already commonly used packages (this is actually something what will just lead to the rare use of BIOdry by people which are more used to use R and adjust data; this is really pity as the package has potential to be widely used).

On the base of this, I recommend to change the structure of the paper and write the paper as manual (see papers describing dplR, detrendeR, TRADER, pointRes and basically all other papers, including your paper describing measuRing). There is always some clear example, which is really missing in your manuscript. Basically, even from the abstract is not clear what is the main purpose of the package. You use multilevel ecological data series, but it is hard to recognize what does it mean. I am really sorry to be so critical, especial as I know that there was for sure a lot of work behind whole package and also manuscript writing, but I feel that in the current stage is not manuscript useful for users and I think it should be the main purpose of this manuscript. I am sure that it will be not so complicated to write the manuscript in easier way and most of the detail precise description of processes behind functions, put to the Appendix. Anyway, I am looking forward to use this package and hope that I will find paper, which will really provide an information which I would like to see as user of package, not the description

of various complicated calculation, which most of the users do not care.

Some more specific comments

1. I cannot agree with second sentence in introduction (Diverse methods and software for measuring Tree Rings in Wood (TRW), analyzing climate data, and compiling statistics are scattered throughout the literature). First of all, what authors mean by "measuring tree rings in wood" (it is already in Abstract)? If you mean for measuring tree-ring width, write it correctly (because this is how abbreviation TRW is always used and, according my opinion, you use it in this meaning in the rest of the manuscript). Secondly, sentence is simply not true as there is variety of software's and literature describing these methods and I do not see point for this criticism. There is a lot of dendrochronological books, which describe in one place what you told that it is missing (e.g. Speer JH (2010) Fundamentals of Tree-ring Research, University of Arizona Press; but plenty of others). There is also a lot of papers which describe the software separately. Simply, there is no reason for this sentence.

2. your package is partly working with another package, bootRes (not clear how, but you wrote it). I think it will be much better if you will make the connection with newer package (basically update of bootRes), treeclim; Zang, C., & Biondi, F. (2015). treeclim: an R package for the numerical calibration of proxy-climate relationships. Ecography, 38(4), 431-436.

3. On Figure 1 is a lot of abbreviations, which are not explained at all, this should be changed. 4. in the R-code 1, you have a mistake, there is one more bracket, should be like this: MoreArgs_td <- list( z = 2003, mp = c(1,1), un = c('mm','cm'))

5. Actually there is an error also in other point:

Error in lme.formula(fixed = log(x) $\sim$ log(csx) + log(time), data = fixed, :

nlminb problem, convergence error code = 1

message = singular convergence (7)

```
diameters <- modelFrame(rd = Prings05, fn = fn_td,

lv = lv_td,

MoreArgs = MoreArgs_td,

y = Pradii03,

form = 'tdForm',

log.t = TRUE,

method = 'REML')
```

6. I think you should simply wrote in the introduction, something like: There are already available packages for basic tree-ring width manipulation (dplR; Bunn 2008), investigation of dendroclimatic relationship (treeclim or bootRes; Zang and Biondi 2013, 2015) as well as assessment of dendroecological processes (TRADER, Altman et al. 2014), while the com plex tool for connection and deep investigation of Multilevel Ecological Data Series (MEDS) is still missing. Thus, you will put your package to the broader content.

7. R-code 2 does not work if I will start to run it, simply because there are some missing steps before and data used in this first step are not available.

References

Altman J, Fibich P, Dolezal J, Aakala T (2014) TRADER: a package for Tree Ring Analysis of Disturbance Events in R. Dendrochronologia, 32, 107-112.

Bunn, A.G., 2008. A dendrochronology program library in R (dplR). Dendrochronologia 26, 115–124. Zang, C., Biondi, F., 2013. Dendroclimatic calibration in R: the bootRes package for response and correlation function analysis. Dendrochronologia 31, 68–74.

Zang, C., & Biondi, F. (2015). treeclim: an R package for the numerical calibration of proxy-climate relationships. Ecography, 38(4), 431-436.

---

## Referee Comment (RC2) · Anonymous Referee #1 · 3 Feb 2017

This paper by Lara et al. is currently somewhat of an chimaera between a software manual and methods paper for a special case of mixed effects modelling. Unfortunately, both parts are unsatisfactory, which makes a review rather tough.

Judging from its title, this paper is intended to introduce a software package to the community (like the papers introducing dplR, TRADER, pointRes, treeclim, etc.). This means, this review should not be about the package's functionality or the quality of the code, but rather about the presentation of the package. The functionality and statistical reasoning behind it is described elsewhere (Lara et al. 2013 Agricultural and Forest Meteorology) and its popularity and benefit can thus be assessed independently.

The functions presented with the package are more on the complex side of things, especially since they overload already complex functionality (nlme-functions) with even

more complexity and paradigms. This calls for a more detailed, example-driven step-by-step guide, where references are made to the appropriate places in the methodological literature, but where the reader is not overwhelmed by statistical reasoning. In this, I very much agree with Reviewer #2. In contrast to Reviewer #2, I could run all example code; but still, I was left somewhat puzzled about what the code actually tries to do. Recommendation: use ecological storylines to tell a story with data, and explain your package alongside. For example, as a reader, the sentence "For instance, form = 'lmeForm' can be implemented to detrend normalized aridity indexes." is most probably not helpful when you do not give an example why would want to do that in the first place. Also, as the authors claim their package to be usable for MEDS in general, an application example using other data than dendro data would be convincing.

A few more specific things to consider:

- TRW is usually not "Tree Rings in Wood" but tree-ring width - the authors' concept of detrending is unclear and possibly varying throughout the manuscript - stable isotopes are more of a *tool* for certain research areas in dendro - use lazy data (see Writing R Extensions or ?utils::data) - don't recommend using 'require()'

---

## Author Comment (AC1) · 10 Apr 2017

We would like to thank the reviewer for his constructive criticism and for the time spent analyzing this manuscript. The responses and explanations to his comments are listed below (the reviewer's comments are in italics):

Issue #1 The present study introduces new R-package which can be useful in tree-ring research, but also in other time-series analyses. The manuscript is clearly written and contains enough detail to follow the analytical protocol employed. Package is already available at CRAN.

R 1.1: We thank the reviewer for his comments about the writing quality of the manuscript.

[Figure]

Issue # 2 However, I feel that such type of manuscript should be mostly dedicated to the description of individual functions and how user can modify this and what will be the output. However, this paper is farfrom clear cookbook for potential package user, which I believe is really pity.

R 1.2: We agree with the reviewer. In order to favor the potential package use we have included the code into the manuscript with clear descriptions of functions, arguments, inputs requirements and function outputs.

Issue #3 There is a lot of space dedicated to explanation of mathematical processes, but this should be rather in the Appendix (honestly, most of the readers and potential users will not care about it, it is your responsibility that used methods are correct, with this description you will not increase the understanding for people which simply do not understand these methods, just to want to use them as a tool without deep knowledge how it works).

R 1.3: We agree with the reviewer . Mathematical processes/formulations have been specified in the Appendix section.

Issue #4 I also found some errors in the script, which is really something what I would not expect from the manuscript describing the package.

R 1.4: We agree with the reviewer that the code should run without errors. We have tested the examples of the manuscript in different machine architectures as Linux (Ubuntu), Macintosh, and Windows. However, we have not found errors in the script. We have programmed BIOdry in S3 language for R version 3.3.2. The package could not work well in preliminary versions of R. Changes in the character encoding of the script could also affect the way that R translates the code. We will continue checking the in-package algorithms in order to detect and fix any error of the code.

Issue #5 There are some Figures in the manuscript, which I suppose are produced by some function of the package. However, this is really not clear and it is only my opinion.

[Figure]
R 1.5: We have included in the examples the code to reproduce figures of the outputs such as dendroclimatic fluctuations and Mantel correlograms.

Issue #6 I would expect that there will be clear line of script shown in manuscript, as basically all packages published in dendrochronology did, and this line will be accompanied by short description what it is doing and if there is a figure produced, then it should be explicitly written.

R 1.6: We have included the code into the manuscript with clear descriptions about function arguments, inputs and outputs.

Issue #7 Authors mention that "the BIOdry package can interact with other dendrochronological libraries in R, including dplR, bootRes, and measuRing", however, they show only one example of communication of dplR package and some connection with other packages is not shown at all.

R 1.7: We feel sorry if we did not explain properly this paragraph. By saying that the package can interact with other dendrochronological libraries we ment that our package processed data in the same way that other dendrochronological packages did. We have improved the writing of this paragraph and adapted the code for the data inputs to standard dendroclimatic formats.

Issue #8 In addition, even this one example is actually quite rather showing that these packages are not able to communicate, as the format of data is different and I do not understand why it is not unified or there is not some function for transformation to commonly used and already commonly used packages (this is actually something what will just lead to the rare use of BIOdry by people which are more used to use R and adjust data; this is really pity as the package has potential to be widely used).

R 1.8: We agree with the reviewer that the package should process standard dendrochronological data formats. We have changed the in-package code so the model-Frame() function can read TRW/climate data in the same way that other dendrochrono-

logical packages do: e.g. TRW chronologies.

Issue #9On the base of this, I recommend to change the structure of the paper and write the paper as manual (see papers describing dplR, detrendeR, TRADER, pointRes and basically all other papers, including your paper describing measuRing). There is always some clear example, which is really missing in your manuscript. Basically, even from the abstract is not clear what the main purpose of the package is.

R 1.9: We agree with the reviewer and we have changed the paper's structure as manual with a clear example workflow. The abstract has also been changed to better resume the package functionalities.

Issue #10 You use multilevel ecological data series, but it is hard to recognize what it means.

R 1.10: We have explained in the manuscript the relevance of processing dendroclimatic data in the form of Multilevel Ecological Data Series (MEDS). These are dendroclimatic data frames containing categorical variables (i.e. ecological factors) which affect the dendroclimatic records. The MEDS are processed by BIOdry to derive dendroclimatic inputs and account for source variability.

R 1.10.1: We agree with the reviewer that package users do not aim to transform their standard-data formats into MEDS. For this reason we have adapted the package so its internal algorithms can automatically format the data.

Issue #11 I am really sorry to be so critical, especial as I know that there was for sure a lot of work behind whole package and also manuscript writing, but I feel that in the current stage is not manuscript useful for users and I think it should be the main purpose of this manuscript.

R 1.11: We have transformed the manuscript into a manual with clear example workflow.

Issue #12 I am sure that it will be not so complicated to write the manuscript in easier

way and most of the detail precise description of processes behind functions, put to the Appendix.

R 1.12: We agree with the reviewer that including the in-package functionalities in the appendix and focusing the paper body in a line code example could be more useful for users of the package. We have developed an appendix section where such functionalities are defined.

Issue #13 Anyway, I am looking forward to use this package and hope that I will find paper, which will really provide an information which I would like to see as user of package, not the description of various complicated calculation, which most of the users do not care.

R 1.13: We would like to thank the reviewer for his interest in the package implementation. We have changed the examples in the manuscript so they can be more useful and understandable.

Issue #14 I cannot agree with second sentence in introduction (Diverse methods and soft- ware for measuring Tree Rings in Wood (TRW), analyzing climate data, and compiling statistics are scattered throughout the literature). First of all, what authors mean by "measuring tree rings in wood" (it is already in Abstract)? If you mean for measuring tree-ring width, write it correctly (because this is how abbreviation TRW is always used and, according my opinion, you use it in this meaning in the rest of the manuscript). Secondly, sentence is simply not true as there is variety of software's and literature describing these methods and I do not see point for this criticism. There are a lot of dendrochronological books, which describe in one place what you told that it is missing (e.g. Speer JH (2010) Fundamentals of Tree-ring Research, University of Arizona Press; but plenty of others). There is also a lot of papers which describe the software separately. Simply, there is no reason for this sentence.

R 1.14: We agree with the reviewer that TRW should be changed to tree-ring width.

R 1.15: We have also changed the sentence about the dendroclimatic methods, as the reviewer suggests in a paragraph below (Issue # 19).

Issue #16 your package is partly working with another package, bootRes (not clear how, but you wrote it). I think it will be much better if you will make the connection with newer package (basically update of bootRes), treeclim; Zang, C., & Biondi, F. (2015). treeclim: an R package for the numerical calibration of proxy-climate relationships. Ecography, 38(4), 431-436.

R 1.16: We have better explained that the data inputs required by BIOdry are similar to the inputs required by other dendrochronological packages, and mentioned dplR, bootRes, and treeclim packages as examples.

Issue #17 On Figure 1 is a lot of abbreviations, which are not explained at all, this should be changed. 4. 4. in the R-code 1, you have a mistake, there is one more bracket, should be like this: MoreArgs_td <- list( z = 2003, mp = c(1,1), un = c('mm','cm'))

R 1.17: We have defined abbreviations in Figure 4.

R: Brackets in the MoreArgs list have been checked and correctly written.

Issue #18 Actually there is an error also in other point: Printer-friendly version Error in lme.formula(fixed = log(x) áĹij log(csx) + log(time), data = fixed, : nlminb problem, convergence error code = 1 message = singular convergence (7)

diameters <- modelFrame(rd = Prings05, fn = fn_td, lv = lv_td, MoreArgs = MoreArgs_td, y = Pradii03, form = 'tdForm', log.t = TRUE, method = 'REML'))

R 1.18: See response R: 1.4.

Issue #19 I think you should simply wrote in the introduction, something like: There are already available packages for basic tree-ring width manipulation (dplR; Bunn 2008), investigation of dendroclimatic relationship (treeclim or bootRes; Zang and Biondi 2013,

2015) as well as assessment of dendroecological processes (TRADER, Altman et al. 2014), while the com plex tool for connection and deep investigation of Multilevel Ecological Data Series (MEDS) is still missing. Thus, you will put your package to the broader content.

R 1.19: We agree with the reviewer and the paragraph he proposed has been included in the manuscript.

Issue #20. R-code 2 does not work if I will start to run it, simply because there are some missing steps before and data used in this first step are not available.

R 1.20: We agree with the reviewer. We have included an example which considers all steps for the R code can work properly.

Issue #21 References

Altman J, Fibich P, Dolezal J, Aakala T (2014) TRADER: a package for Tree Ring Analysis of Disturbance Events in R. Dendrochronologia, 32, 107-112.

Bunn, A.G., 2008. A dendrochronology program library in R (dplR). Dendrochronologia 26, 115–124.

Zang, C., Biondi, F., 2013. Dendroclimatic calibration in R: the bootRes package for response and correlation function analysis. Dendrochronologia 31, 68–74.

Zang, C., & Biondi, F. (2015). treeclim: an R package for the numerical calibration of proxy-climate relationships. Ecography, 38(4), 431-436.

R 1.21: We have included the new bibliography suggested by the reviewer.

---

## Author Comment (AC2) · 10 Apr 2017

We would like to thank the reviewer for careful and thorough reading of this manuscript and for the thoughtful comments and constructive suggestions, which helped us to improve the quality of this manuscript. Our response follows (the reviewer's comments are in italics).

Issue #1 This paper by Lara et al. is currently somewhat of an chimaera between a software manual and methods paper for a special case of mixed effects modelling. Unfortunately, both parts are unsatisfactory, which makes a review rather tough.

Judging from its title, this paper is intended to introduce a software package to the community (like the papers introducing dplR, TRADER, pointRes, treeclim, etc.). This means, this review should not be about the package's functionality or the quality of the

code, but rather about the presentation of the package. The functionality and statistical reasoning behind it is described elsewhere (Lara et al. 2013 Agricultural and Forest Meteorology) and its popularity and benefit can thus be assessed independently.

The functions presented with the package are more on the complex side of things, especially since they overload already complex functionality (nlme-functions) with even more complexity and paradigms. This calls for a more detailed, example-driven step-by-step guide, where references are made to the appropriate places in the method-ological literature, but where the reader is not overwhelmed by statistical reasoning. In this, I very much agree with Reviewer #2.

R 2.1: As a response to the reviewer's comments we have focused the paper in a step-by- step guide while the in-package routines were explained in the appendix.

Issue #2 In contrast to Reviewer #2, I could run all example code; but still, I was left somewhat puzzled about what the code actually tries to do.

R 2.2: The new line code example clearly explains formulation of arguments, inputs and outputs. The R-code has been tested in machines with Linux, Mackintosh, and Windows.

Issue #3 Recommendation: use ecological storylines to tell a story with data, and explain your package alongside. For example, as a reader, the sentence "For instance, form = 'lmeForm' can be implemented to detrend normalized aridity indexes." is most probably not helpful when you do not give an example why would want to do that in the first place.

R 2.3: The recommendation of the reviewer is an excellent idea. A storyline figure to illustrate principal objectives and steps of the example has been included in the manuscript.

Also, as the authors claim their package to be usable for MEDS in general, an application example using other data than dendro data would be convincing.

R 2.3: Considering the algorithm's complexity, we propose to focus the paper on one specific example to model dendroclimatic data and to avoid including other ecological implementations.

Issue #4 A few more specific things to consider:- TRW is usually not "Tree Rings in Wood" but tree-ring width

R 2.4: We agree with the reviewer that TRW should be changed to tree-ring width.

Issue #5 the authors' concept of detrending is unclear and possibly varying throughout the manuscript

R 2.5: We can better explain in the manuscript that the detrending process means subtraction of trends in dendroclimatic data by fitting lme regression.

Issue #6 stable isotopes are more of a *tool* for certain research areas in dendro

R 2.6: We could be more specific with the concept of stable isotopes as a tool for certain areas in dendrochronology.

Issue #7 use lazy data (see Writing R Extensions or ?utils::data)

R 2.7: We are going to use lazy data for package implementation. This will make the R session start up faster and use less virtual memory.

Issue #8 don't recommend using 'require()'

R 2.8: We agree with the reviewer. We will use library() instead of require() to load the package into the R environment.